# PeerJ

# Sampling locality is more detectable than taxonomy or ecology in the gut microbiota of the brood-parasitic Brown-headed Cowbird (*Molothrus ater*)

Sarah M. Hird[1,2], Bryan C. Carstens[3], Steven W. Cardiff[2], Donna L. Dittmann[2] and Robb T. Brumfield[1,2]

[1] Department of Biological Sciences, Louisiana State University, Baton Rouge, LA, USA
[2] Museum of Natural Science, Louisiana State University, Baton Rouge, LA, USA
[3] Department of Evolution, Ecology and Organismal Biology, Ohio State University, Columbus, OH, USA

Corresponding author
Sarah M. Hird,
sarah.hird@gmail.com

## ABSTRACT

Brown-headed Cowbirds (*Molothrus ater*) are the most widespread avian brood parasite in North America, laying their eggs in the nests of approximately 250 host species that raise the cowbird nestlings as their own. It is currently unknown how these heterospecific hosts influence the cowbird gut microbiota relative to other factors, such as the local environment and genetics. We test a Nature Hypothesis (positing the importance of cowbird genetics) and a Nurture Hypothesis (where the host parents are most influential to cowbird gut microbiota) using the V6 region of 16S rRNA as a microbial fingerprint of the gut from 32 cowbird samples and 16 potential hosts from nine species. We test additional hypotheses regarding the influence of the local environment and age of the birds. We found no evidence for the Nature Hypothesis and little support for the Nurture Hypothesis. Cowbird gut microbiota did not form a clade, but neither did members of the host species. Rather, the physical location, diet and age of the bird, whether cowbird or host, were the most significant categorical variables. Thus, passerine gut microbiota may be most strongly influenced by environmental factors. To put this variation in a broader context, we compared the bird data to a fecal microbiota dataset of 38 mammal species and 22 insect species. Insects were always the most variable; on some axes, we found more variation within cowbirds than across all mammals. Taken together, passerine gut microbiota may be more variable and environmentally determined than other taxonomic groups examined to date.

## INTRODUCTION

Birds interact mutualistically with microorganisms on a multitude of axes across anatomical and functional systems, from maintenance of their feathers (*Ruiz-Rodriguez et al., 2009a*) to parenting behavior (*Cook et al., 2005*). Microbes that inhabit the gut are particularly important (*Kohl, 2012*) because they have been shown to affect a number

of processes, including brain development (*Heijtz et al., 2011*), mate choice (*Sharon et al., 2010*), hybrid speciation (*Brucker & Bordenstein, 2013*) and an organism's ability to obtain essential amino acids (*Gündüz & Douglas, 2009*). The number of genes in the gut microbiota can outnumber host genes by several orders of magnitude (*Bäckhed et al., 2005*). These genes comprise a vast genetic resource that shares many of the evolutionary pressures of the host yet are also exposed to local selection pressures within the gut (*Ley, Peterson & Gordon, 2006*). Despite this biological importance, it is not well understood how the microbiota is assembled.

As in other vertebrates, the gut microbiota of birds are influenced by both genetic and non-genetic factors. *Banks, Cary & Hogg (2009)* found genetic distance between conspecific Adelie penguin individuals (*Pygoscelis adeliae*) was the most significant correlate to fecal microbiota, with no correlation to physical distance. *Dewar et al. (2013)* found that the major phyla in the feces of four species of penguin clustered by taxonomic group. However, they noted some overlap between the species, and the underlying cause of the clustering was not determined. Other studies have found that non-genetic factors, such as geography and age, are most important in structuring the microbiota. *Klomp et al. (2008)* used five bacterial taxa in a discriminant function analysis to correctly assign geographic origin to Spotted Towhee individuals (*Pipilo maculatus*). Hoatzins (*Opisthocomus hoazin*) exhibit population level differentiation in their crop microbiota, although this may be attributable to diet, local environment or genetics (*Godoy-Vitorino et al., 2012*). Hoatzins also show age related effects on their crop microbiota (*Godoy-Vitorino et al., 2010*), which has also been found in Black-legged Kittiwakes, *Rissa tridactyla* (*van Dongen et al., 2013*).

Avian brood parasites offer a unique natural system to investigate these processes because vertical (genetic or phylogenetic) and horizontal (environmental or ecological) transmission of microbes are naturally decoupled. Instead of building nests and raising their young, brood parasites lay their eggs in the nests of suitable brood hosts, thus leaving brood hosts to invest reproductive resources in the parasitic young instead of their own. Direct transmission of gut microbes from the brood host to the brood parasite is possible because altricial bird species feed their nestling and fledgling young by placing food directly in the mouth of young, either from their bills or by regurgitating from their crops (which the young eat out of the parent's mouth). A study of brood parasitic cuckoos (Cuculidae) found significant differentiation of the gut microbiota between the brood parasite young and their nest mates (*Ruiz-Rodriguez et al., 2009b*), indicating that the microbiota of young is inherited from their biological parents. In contrast, a cross-fostering experiment of two tit species (Parulidae), neither of which are brood parasites, found that heterospecific young raised within the same nest had more similar microbiota than was found between biological siblings raised in separate nests (*Lucas & Heeb, 2005*).

Here, we present the first comparative study of gut microbiota in the Brown-headed Cowbirds (*Molothrus ater*, hereinafter cowbirds), a generalist brood-parasitic species in the order Passeriformes. Without any particular egg mimicry, cowbirds parasitize

approximately 250 passerine species in North America (*Lowther, 1993*; *Lowther, 2013*). They have evolved many adaptations for this lifestyle. An unusually good immune system prepares them for development in a variety of environments (*Hahn & Smith, 2011*; *Hahn & Reisen, 2011*), flexibility in egg laying behavior gives the cowbird control over when and where to lay eggs (*Woolfenden et al., 2003*), tolerance of host nestlings increases host parent feeding (*Kilner, Madden & Hauber, 2004*), a relatively large gape width and quick growth rate allow them to outcompete nest mates for resources (*Ortega & Cruz, 1992*) and thick eggshells protect the egg from puncture ejection by brood hosts (*Spaw & Rohwer, 1987*).

Our primary goal was to evaluate several hypotheses concerning how the cowbird gut microbiota is structured, using data from high-throughput sequencing of a single "universal" bacterial marker (*Clarridge III, 2004*) in 32 cowbirds and individuals from 9 species parasitized by cowbirds. Because few comparative gut microbiota studies have included birds, we put the variation contained within our samples in the larger context of mammal and insect guts as well.

Importantly, the samples used in this study were not collected with this particular study in mind; a secondary goal of the study was to assess to what extent microbial data can be gathered from museum specimens and then used to address novel questions. Given the recent interest in the importance of microbiota on host organisms' general health (*Claesson et al., 2012*; *Clemente et al., 2012*; *Sekirov et al., 2010*) and evolution (*Brucker & Bordenstein, 2013*; *Zilber-Rosenberg & Rosenberg, 2008*; *Sharon et al., 2010*), suitably collected and preserved museum specimens could provide a novel source of samples for investigators.

## Hypotheses

The four hypotheses we evaluated concerning the structuring of cowbird microbiota are not mutually exclusive. The Nature Hypothesis posits that cowbird gut microbiota are determined predominantly by their biological cowbird parents (Fig. 1A), with the brood host contributing little to the microbiota of the cowbird young. The Nature Hypothesis would be corroborated if we found that the gut microbiota of cowbird samples are most similar to those of other cowbirds, rather than to the microbiota of brood host species. Under the alternative Nurture Hypothesis, the brood host exerts the most influence on its brood's gut microbiota, regardless of its genetic relationship to them (Fig. 1B). This hypothesis predicts that the microbiota of each cowbird should be more similar to that of its brood host species than to other cowbirds. The Nurture Hypothesis would be supported if clusters of cowbird and brood host individuals having similar gut microbiota are detected.

The Environment Hypothesis (Fig. 1C) posits that the local environment (e.g., climate, flora, fauna, etc.) accounts for similarity of gut microbiota. It predicts that the microbiota of birds in closer geographic proximity will be most similar, despite their genetic background, ecology or evolutionary history. Finally, cowbirds may have different gut microbiota assemblages at different life stages (the Convergence Hypothesis, Fig. 1D). One prediction is that younger cowbirds have a more diverse microbiota assemblage that is able to utilize a variety of diets but which converges to a stable cowbird-like microbiota

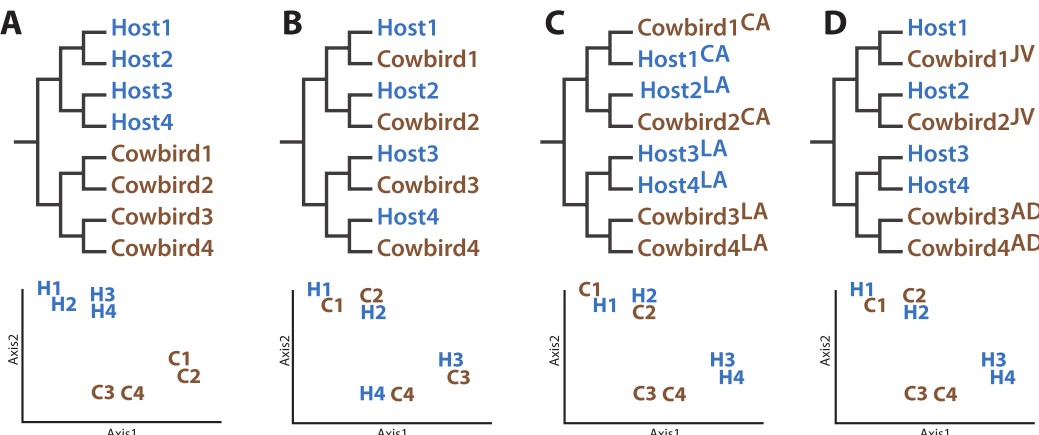

**Figure 1 Diagrams of how the data could support the major hypotheses.** Diagrams of how the data could support the major hypotheses through dendrograms (top row) and principle coordinates analyses (bottom row), using four (fictional) cowbird microbiota samples (C01–C04) and four host species (H01–H04). (A) Cowbird microbiota may be most closely related to other cowbirds, despite their host species, the "Nature Hypothesis". (B) Cowbird microbiota may most closely resemble their host microbiota, the "Nurture Hypothesis". (C) Local factors may determine the gut microbiota, causing birds from different localities (LA vs. CA) to be most similar. (D) Cowbird microbiota may shift between being host-like when they are juveniles (JV) and cowbird-like when they are mature (AD), the "Convergence Hypothesis".

as the cowbirds mature. This will be supported if the microbiota of adult cowbirds form their own clade (to the exclusion of the juveniles) or if the gut microbiota of juveniles are taxonomically more diverse than adults. ("Juvenile" is defined here as an individual having the combination of juvenile plumage, a completely unossified cranium, presence of a Bursa of Fabricius and undeveloped gonads.)

## MATERIALS AND METHODS

### Sampling

We sampled from birds available in the LSU Museum of Natural Science's Collection of Genetic Resources for this investigation, including 32 cowbirds (Icteridae: *Molothrus ater*) from Louisiana (LA, near Baton Rouge) and California (CA, near Weldon) and 16 individuals from nine known parasitized passerine species, all but one from LA (Table 1, more detailed specimen information in Table S1): Northern Cardinal (Cardinalidae: *Cardinalis cardinalis*), House Finch (Fringillidae: *Haemorhous mexicanus*), Orchard Oriole (Icteridae: *Icterus spurius*), Indigo Bunting (Cardinalidae: *Passerina cyanea*), Blue-gray Gnatcatcher (Polioptilidae: *Polioptila caerulea*), Prothonotary Warbler (Parulidae: *Protonotaria citrea*), Carolina Wren (Troglodytidae: *Thryothorus ludovicianus*), White-eyed Vireo (Vireonidae: *Vireo griseus*), Hooded Warbler (Parulidae: *Setophaga citrina*). Birds were collected under a general collection permit for another project (USFWS scientific collecting permit MB679782; LDFW permit LNHP-13-032) and frozen within two hours of their collection. Throughout the manuscript, individuals are identified by their common names and an individual identifying number. One individual (NorthernCardinal4) served as a replicate

**Table 1 Sample and species information.** Number of individuals from each locality, adult (AD)/juvenile (JV) status and rate of Brown-headed cowbird parasitism (*Ortega, 1998*) for each species.

| | California | | Louisiana | | Totals | Parasitism rate (%) |
|---|---|---|---|---|---|---|
| | **JV** | **AD** | **JV** | **AD** | | |
| Northern Cardinal | 0 | 0 | 2 | 2 | 4 | 2.7–100 |
| House Finch | 1 | 0 | 0 | 0 | 1 | 0–58.3 |
| Orchard Oriole | 0 | 0 | 0 | 2 | 2 | 6.7–100 |
| Brown-headed Cowbird | 11 | 1 | 8 | 12 | 32 | N/A |
| Indigo Bunting | 0 | 0 | 0 | 1 | 1 | 0–71.4 |
| Blue-gray Gnatcatcher | 0 | 0 | 2 | 0 | 2 | 0–75.9 |
| Prothonotary Warbler | 0 | 0 | 1 | 1 | 2 | 6.7–20.9 |
| Carolina Wren | 0 | 0 | 1 | 1 | 2 | 0–33 |
| White-eyed Vireo | 0 | 0 | 0 | 1 | 1 | 40 |
| Hooded Warbler | 0 | 0 | 0 | 1 | 1 | No data |
| Totals | 12 | 1 | 14 | 21 | | |

to assess PCR/sequencing bias; these samples are identified as NorthernCardinal4.1 and NorthernCardinal4.2 and bring the total number of samples to 49.

## DNA extraction, amplification, sequencing and quality control

The entire digestive tract was removed when the bird was thawed for museum specimen preparation. Total DNA was immediately extracted from the contents of the large intestine, halfway between the ceca and cloaca, using a QIAamp DNA Stool Mini Kit (QIAGEN). Following *Gloor et al. (2010)*, we used combinatoric primers and massive multiplexing of PCR amplicons for sequencing on one lane of an Illumina Hi-Seq. This method uses paired-end sequencing technology to generate pairs of sequences with 100% overlap across variable region 6 (V6) of the 16S component of rRNA; primer sequences align to positions 967–985 and 1078–1061 on *Escherichia coli* 16S rRNA (*Gloor et al., 2010*).

We used several measures of sequence quality control. First, both reads of a given pair had to match across 100% of the bases. The pairs also must have no errors in tag sequence or priming sequence. We used the BELLEROPHON (*Huber, Faulkner & Hugenholtz, 2004*) function within the MOTHUR program (*Schloss et al., 2009*) to identify and discard potentially chimeric sequences. Finally, we used MOTHUR to discard sequences that did not blast to the domain Bacteria. The reads passing these filters were included in the final dataset.

## Clustering analyses

Individuals were partitioned into four datasets in order to test the hypotheses outlined above: *All Birds* ($N = 49$), *All Louisiana Birds* ($N = 35$), *Cowbirds Only* ($N = 32$), *Hosts Only* ($N = 17$).

The microbial ecology package QIIME (*Caporaso et al., 2010*) was used for the following analyses. First, the reads were assigned to phylotypes at 97% sequence similarity because

3% is frequently cited as the "species" level of microbial taxonomy (*Schloss & Handelsman, 2005*). Next, we assigned taxonomies to OTUs using the RDP CLASSIFIER PROGRAM (*Wang et al., 2007*), with the default confidence threshold of 80%. A pairwise matrix of distances between each gut microbial community (i.e., each bird specimen) was constructed using UniFrac (*Lozupone & Knight, 2005*). UniFrac distances are calculated based on the amount of branch length in a phylogenetic tree that is unique to either of two environments (versus how much of the tree is shared by the environments). These distances can be weighted by abundance or be based on presence–absence of OTUs (unweighted); we use both distance matrices for analyses because it is unclear which method is best for representing gut microbiota samples and is more transparent than choosing one arbitrarily. Our microbial phylogenetic tree was constructed with FASTTREE (*Price, Dehal & Arkin, 2009*). To reduce the effects of sampling (sequencing) bias, all individuals were randomly reduced to 5 018 reads, equal to the lowest number of reads for any bird in the dataset.

We constructed UPGMA dendrograms based on both the unweighted and weighted UniFrac distances to visually represent the relatedness of the gut microbiota for all five datasets and test the hypotheses. As a confidence metric, we jackknifed the trees using the QIIME recommendation of 75% of the reads used in the rarefaction (3 760) with 10 replicates. Principal coordinates analysis (PCoA) was also performed on both the weighted and unweighted UniFrac distance matrices.

As a complement to the phylogenetic-based methods, we visualized the data with nonmetric multidimensional scaling (NMDS). We square root-transformed the percentage of each sample that belonged to each bacterial phylum, then created a pairwise distance matrix using Bray–Curtis dissimilarity, applied through the *vegdist* function of the VEGAN package (*Oksanen et al., 2011*) in R (*R Development Core Team, 2010*). The *nmds* function of the ECODIST package (*Goslee & Urban, 2007*) was then used to calculate the two-dimensional positions of the samples (such that closer samples are more similar), the stress and $R^2$ value of the plot. Stress values $>0.3$ should not be considered valid whereas values $<0.2$ can be considered a good representation of the data (*Quinn & Keough, 2002*).

To specifically test the Convergence Hypothesis, we compared UniFrac distances between and within adult cowbirds, juvenile cowbirds and brood hosts. If adult cowbirds converge on a cowbird-specific microbiota, adults will have lower pairwise distances than within juveniles or either category to brood hosts. Both weighted and unweighted UniFrac distances were assessed.

## Categorical variable significance

To further explore the factors correlated with gut microbiota, we tested for statistical associations between categorical metadata associated with each bird and the UniFrac distance matrices (Table S1); we used the statistical tools Adonis (*McArdle & Anderson, 2001*) and Anosim (*Clarke, 1993*) implemented in QIIME. The categorical variables included family, genus, species, age (either "juvenile" or "adult" based on percent of skull ossification), locality (LA or CA), diet (the known dietary specialization of the

species: mostly plant material, mostly animal material, both animal and plant material) and stomach contents (what was actually found in the stomach, e.g., "insects" or "white millet"). We also tested the total number of bacterial phyla detected per bird (richness) to see if the phylum-level diversity of the established microbial community was associated with the communities. We calculated significance of all variables for both the weighted and unweighted UniFrac distance matrices with 999 iterations; we also repeated analyses with a higher number of sequences (and fewer individuals) for each dataset to see if the signal changed when more data were included. Datasets were rarefied to 17 000 and 42 000 sequences, reducing the number of birds to 46 and 35, respectively.

Based on results, we ran additional Adonis tests to partition the variation in the samples as due to the age and locality variables. We used the *All Birds* dataset and analyzed the weighted UniFrac distance matrix, unweighted UniFrac distance matrix and a sites (birds) by species (bacterial phyla) matrix, where cells were assigned the value of the number of sequences belonging to each phylum for each bird. We used the *adonis* function of the VEGAN package in R and performed 999 iterations, constraining resampling to be within species. Since the order of variables being tested matters, we tried age then locality as well as locality then age for each of the datasets.

## Comparison to mammals and insects

To put avian gut microbiota in a broader context, we compared our results to a mammal dataset (*Ley et al., 2008*) containing 56 individuals from 56 species across 13 orders (Table S1) and an insect meta-analysis dataset containing 85 individuals from 62 species across seven orders (*Colman, Toolson & Takacs-Vesbach, 2012*). Although the mammal and insect datasets were collected with different methods than those outlined above, most sequence fragments contained the V6 region. We pruned all reads to the same homologous region and length for analysis. We only analyzed samples with greater than 200 sequences. To increase coverage for some species we combined mammals belonging to the same species into single samples. This treatment should not skew the results of our analysis, because *Ley et al. (2008)* found that individuals from the same species clustered together. We taxonomically assigned reads using RDP CLASSIFIER PROGRAM. For PCoA, we rarefied all samples to 200 reads and used the unweighted UniFrac distances as input. We also performed NMDS on samples, as described above. To test for significant associations between class, order and diet categories (herbivore, carnivore, omnivore), we tested each variable against both the weighted and unweighted UniFrac distance matrices in the same manner as above.

## RESULTS

Initial quality control steps resulted in 3 500 665 pairs of reads with no errors in priming sequence, region of overlap or individual tags. Three hundred and thirty three potentially chimeric sequences (0.01% of reads) and 62 201 sequences that did not align to the domain Bacteria (1.7% of reads) were removed. The reads passing these filters were included in the final dataset, totaling 3 438 131 sequences and averaging 70 165 sequences per individual, but reads/sample varied by two orders of magnitude (range: 5018–629 093).

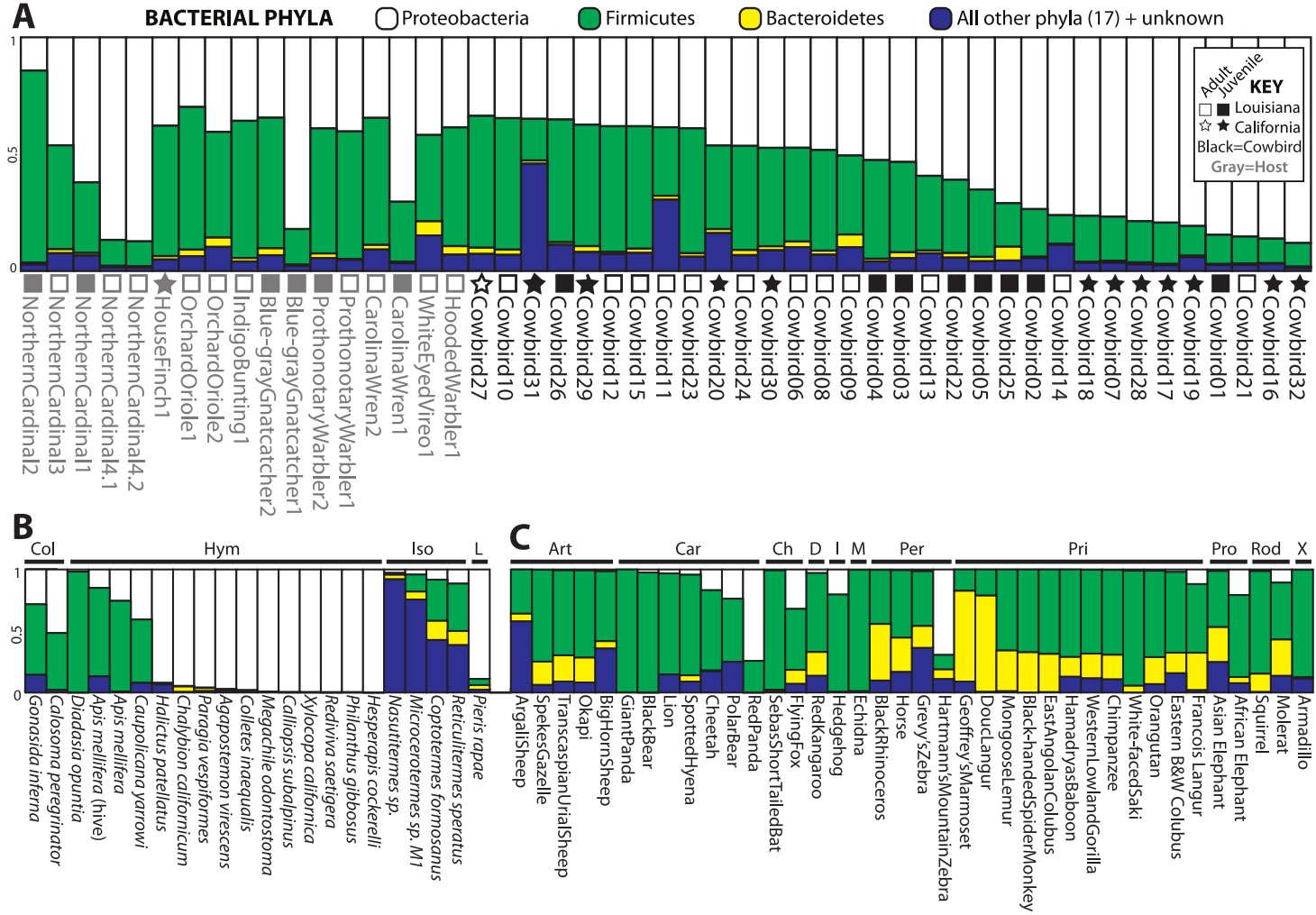

**Figure 2** Relative abundance of the top three bacterial phyla in each sample for **(A) birds, (B) insects and (C) mammals with greater than 200 reads.** Locality and adult/juvenile status are shown for each bird. Insect and mammal orders are depicted by bars across the top of their graphs. Insect orders: COLeoptera, HYMenoptera, ISOptera, Lepidoptera. Mammal orders: ARTiodactyla, CARnivora, CHiroptera, Diprotodontia, Insectivora, Monotremata, PERissodactyla, PRImates, PROboscidea, RODentia, Xenarthra.

Four bacterial phyla were detected in all individuals: Proteobacteria, Firmicutes, Bacteroidetes and Actinobacteria. Proteobacteria and Firmicutes dominated most of the samples (Fig. 2). Proteobacteria constituted an average of 54.7% of sequence reads for an individual, Firmicutes an average of 36.0%, and Actinobacteria and Bacteroidetes an average of 1.3% and 1.7%, respectively. An additional 16 phyla were identified: Acidobacteria, Chloroflexi, Cyanobacteria, Deinococcus-Thermus, Fusobacteria, Gemmatimonadetes, Nitrospira, OD1, OP10, OP11, Planctomycetes, Spirochaetes, TM7, Tenericutes, Thermotogae, Verrucomicrobia. 5.8% of sequences were from unknown phyla within Bacteria. All birds shared 36 genera (Table S1) out of 445 (8%) identified; an additional 139 genus-level OTUs did not align to known genera.

**Table 2** $R^2$ **values of Adonis test for significance of locality and age.** $R^2$ values of Adonis test for significance of locality and age across the weighted and unweighted UniFrac distance matrices and the raw sites by species (birds by bacterial phyla) matrix. Asterisks indicate $p$-values. Since the order of variables matters, two tests were conducted where each variable was ordered first.

|  | Weighted | Unweighted | Counts |
|---|---|---|---|
| Locality | 0.066[**] | 0.016 | 0.027 |
| Age | 0.212[***] | 0.116[*] | 0.147[*] |
| Age | 0.249[***] | 0.115[*] | 0.169[**] |
| Locality | 0.029 | 0.016 | 0.005 |

**Notes.**
[***] <0.01.
[**] <0.05.
[*] <0.10.

## Clustering analyses

Cowbird samples did not cluster together in the UPGMA dendrogram of *All Birds* (Fig. 3). Brood host species having more than one individual also did not cluster together, even when cowbird samples were excluded from the analyses (Fig. S1). NMDS representation of *All Birds* showed little segregation by age or locality (Fig. 3), although the stress of the plot was low (0.1395). In general, UPGMA dendrograms and NMDS plots of all datasets showed little clustering by bird species and high levels of variation (Fig. S1). The two replicate samples, NorthernCardinal4.1 and NorthernCardinal4.2 were most closely related to each other in every analysis.

Pairwise distances were assessed between and within adult cowbirds, juvenile cowbirds and brood hosts to test whether adult cowbirds converged on a cowbird specific microbiota (i.e., if adult cowbird microbiota were more similar to each other than they were to other group comparisons or other groups were to themselves). All pairwise comparisons between and within groups had largely overlapping distributions (Fig. S2) for both weighted and unweighted UniFrac distances, and thus, adults were not more similar to each other than other comparisons.

## Categorical variable significance

There were a total of 12 statistical tests computed for each categorical variable for each dataset (Fig. 4E). Figure 4 shows how frequently significant each of the variables was in each dataset; generally speaking, locality was most frequently significant, followed by age and then diet. Taxonomic categories, stomach contents and bacterial richness were generally not significant.

The multifactorial Adonis tests to assess locality and age were run twice, varying the order of the variables, since this can have an affect on the results on three distance matrices and we had no a priori reason to prefer one variable as being more important than the other. Age was significant in all six tests and locality was significant in one of six tests (Table 2).

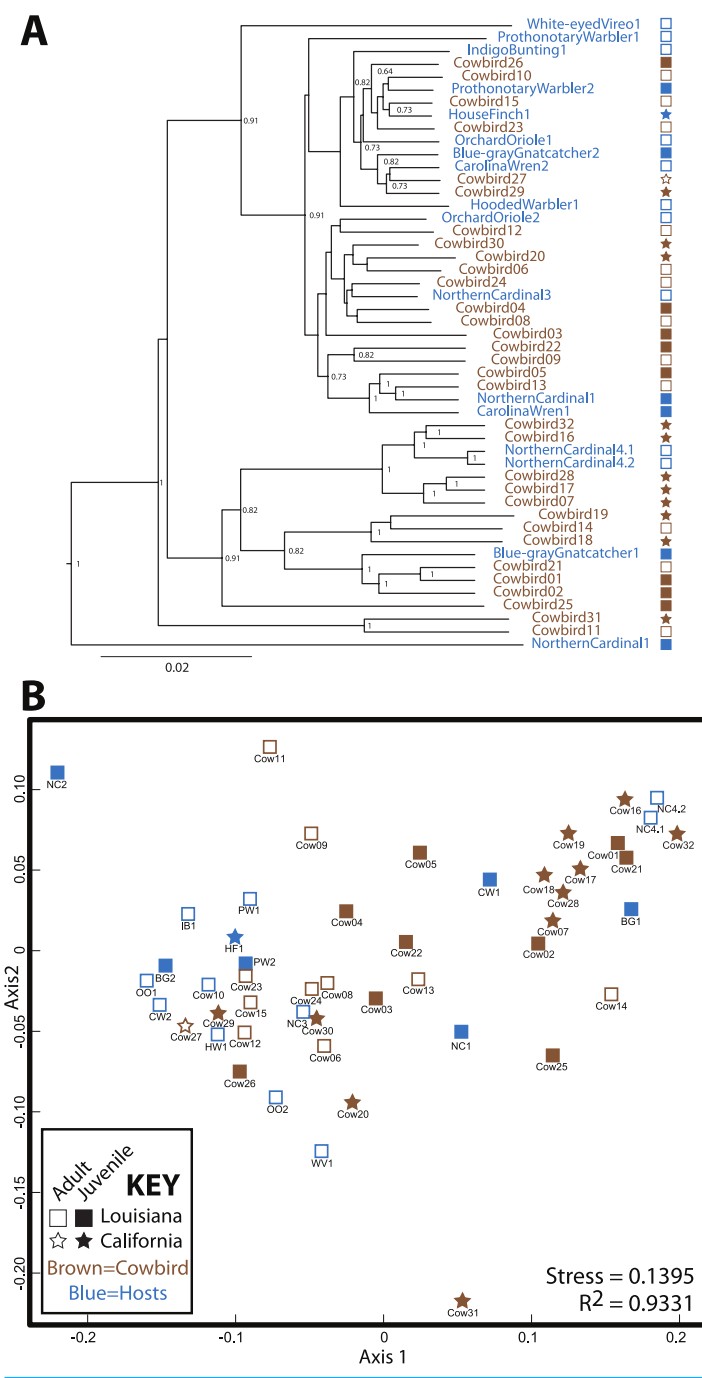

**Figure 3  Relatedness of gut microbiota communities.** Sampling locality is more detectable than taxonomy or ecology in the gut microbiota of the brood-parasitic Brown-Headed Cowbird (*Molothrus ater*) (A) Dendrogram of gut microbiota relatedness based on weighted UniFrac distances; all samples rarefied to 5 018 reads; jackknifed support values are shown for nodes where support >0.70. (B) NMDS ordination of Bray–Curtis dissimilarities of microbiota composition (see Methods).

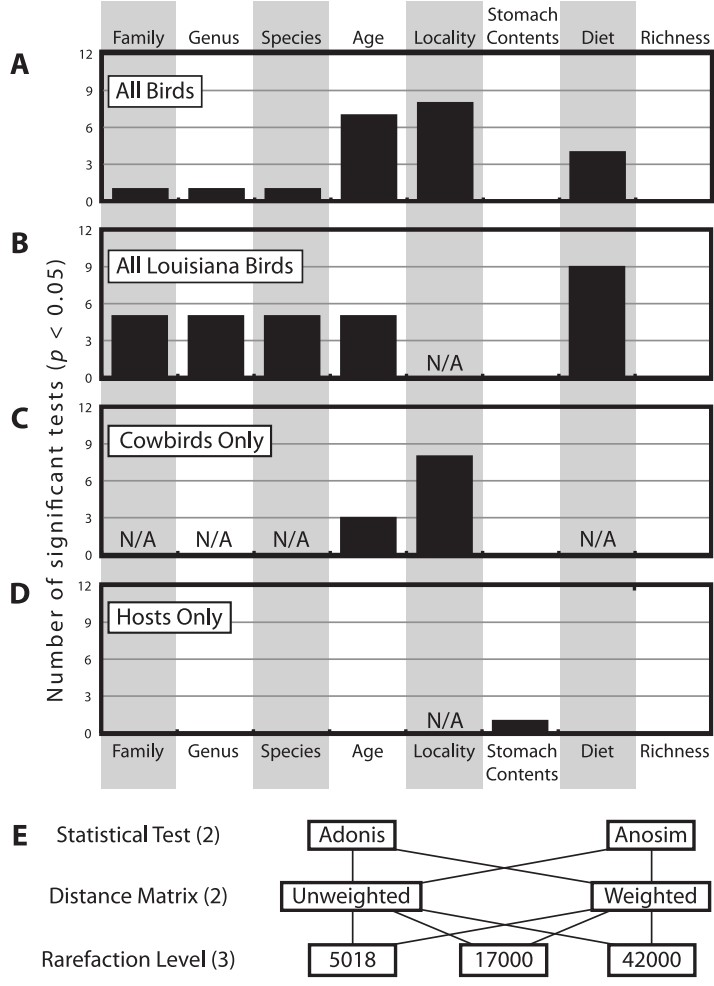

**Figure 4 Histogram of how many times each of the categorical variables was significant.** Histogram of how many times each of the categorical variables was significant at the $p < 0.05$ level (out of 12 total tests, see Methods), for (A) *All Birds*, (B) *All Louisiana Birds,* (C) *Cowbirds Only* and (D) *Hosts Only*. Note some categorical variables were not applicable to some datasets (due to one or fewer individuals belonging to one or more categories). (E) The twelve test contained all permutations of two statistical tests, two distance matrices and three rarefaction levels.

## Comparison to mammals and insects

For all analyses, we only included individuals with more than 200 sequences; for the PCoA, we randomly pruned all individuals to 200 sequences. This reduced the mammal dataset to 38 samples belonging to 11 orders and the insect dataset to 22 individuals belonging to four orders (Table S1). Consistent with *Ley et al. (2008)*, mammals were dominated by Firmicutes and Bacteroidetes (Fig. 2C), whereas bird samples were predominately Firmicutes, with some samples having mostly Proteobacteria or Bacteroidetes (Fig. 2A). Insects also generally contained a majority of Firmicutes although individual samples varied between 100% Proteobacteria and 98% Firmicutes (Fig. 2B). The PCoA showed birds as distinct from mammals and insects, which largely clustered into their respective groups but contained some overlap (Fig. 5A); birds spanned a greater portion of PC2 than

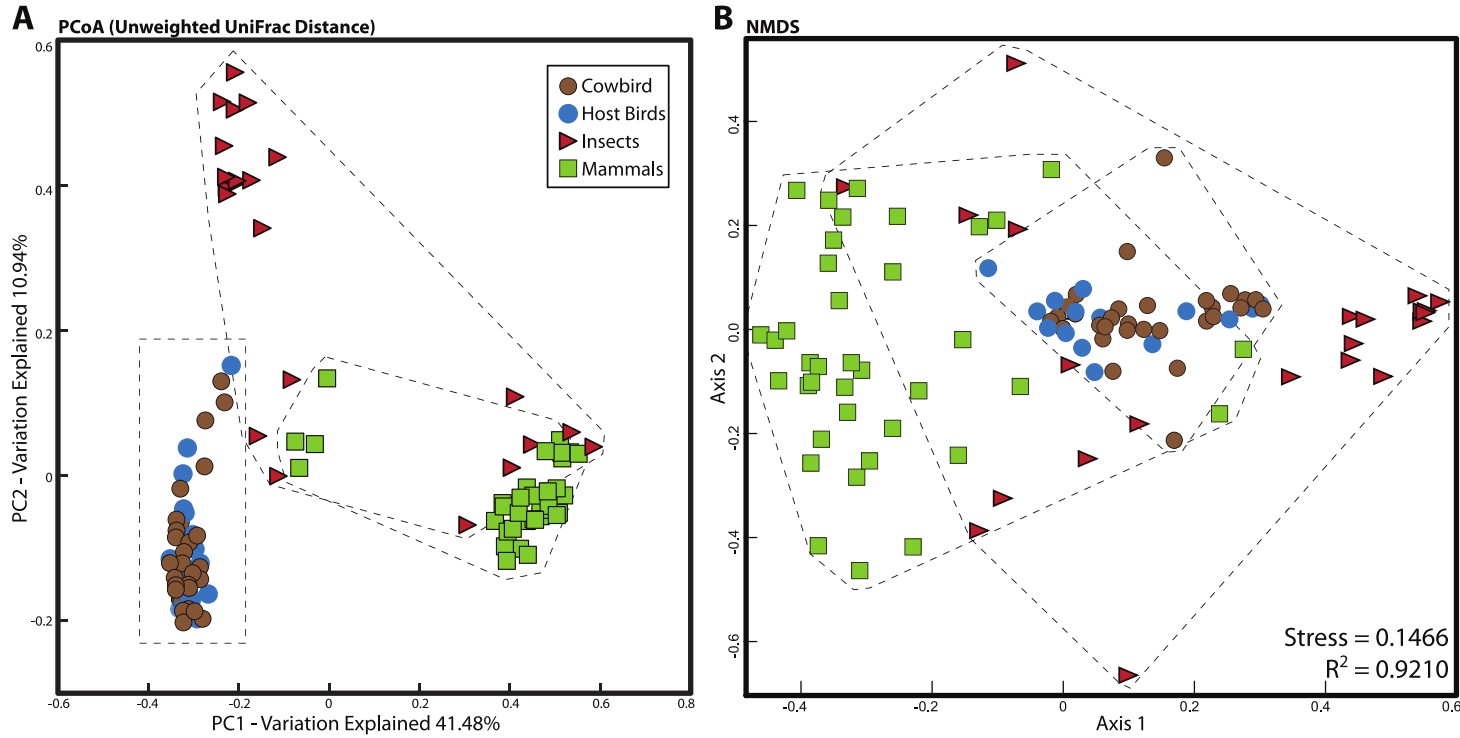

**Figure 5 Relationship between mammal, insect and bird microbiota.** (A) Principal coordinates analysis of unweighted UniFrac distances. (B) Nonmetric multidimensional scaling, using relative abundance of bacterial phyla as input (see Methods). Birds are denoted by circles (cowbirds are brown, brood hosts are blue), mammals are squares and insects are triangles. Dashed lines encapsulate all birds, all insects and all mammals.

all Mammals. NMDS was broadly overlapping but birds clustered together in the middle of the plot (Fig. 5B). Independent Adonis tests for significance of class, order and diet categories revealed highly significant associations ($p < 0.01$) between the gut microbiota and all three variables (Table 3), except for diet and the weighted UniFrac distance matrix, which was not significant ($p = 0.206$).

## DISCUSSION

The microorganisms that are commensal with vertebrates provide essential functions for the host, yet how these complex communities arise and remain stable is largely unknown. Many factors are important, including genetics (*Benson et al., 2010*; *Zoetendal et al., 2001*) and phylogeny (*Ochman et al., 2010*) and non-genetic factors, like ecology (*Muegge et al., 2011*), parental care (*Abecia et al., 2007*; *Kyle & Kyle, 1993*) and the environment (*Bailey et al., 2010*; *Godoy-Vitorino et al., 2012*). The specific importance of these factors in structuring avian microbiota is largely unknown. Because brood parasites are never raised by their genetic parents, they provide a natural system to tease apart genetic and non-genetic influences on the gut microbiota.

Our results do not support a strong role for avian genetics on the structuring of bird gut microbial communities. Our expectation was that if cowbird genetics were most important for structuring the microbial assemblages, the cowbird gut microbial communities would

**Table 3** $R^2$ values of independent Adonis tests for significance of class, taxonomic order and diet. $R^2$ values of independent Adonis tests for significance of class (mammal, insect, bird), taxonomic order and diet (HCO; herbivore, omnivore, carnivore) for the weighted and unweighted UniFrac distance matrices. Asterisks indicate *p*-values.

|  | Weighted | Unweighted |
| --- | --- | --- |
| Class | 0.139[***] | 0.418[***] |
| Order | 0.897[***] | 0.544[***] |
| HCO | 0.029 | 0.080[***] |

**Notes.**
[***] <0.01.

be most similar to each other (see Fig. 1), yet the cowbird gut samples fail to cluster together in any analysis (Fig. 3, Fig. S1). Additionally, individuals from the parasitized bird species are dispersed across the PCAs and cladograms (Fig. 3, Fig. S1), indicating that even in host species, genetics may play little role in the gut microbiota. Tests on the significance of categorical metadata associated with the samples indicate that taxonomic categories are not strongly associated with the gut communities.

Alternatively, several non-genetic factors are correlated with gut microbiota similarity, including locality, age and diet. Sampling locality is the most frequently significant correlate (Fig. 4). Although distinct Louisiana and California clusters were not observed in our analyses, we did identify some locality specific groups—e.g., five juvenile cowbirds from California always grouped together (Cowbirds 7, 16, 17, 28, 32; Fig. 3, Fig. S1). The repeated clustering of these individuals may be the reason that locality and age were both correlated with gut microbiota, instead of one emerging as more important. Both population level (*Godoy-Vitorino et al., 2012*) and age-related (*van Dongen et al., 2013*) differentiation of gut microbiota have been reported in birds (as well as other taxonomic groups (*Vaishampayan et al., 2010*; *Lee et al., 2011*), either or both may explain the results.

It is important to note that we cannot ascribe the differences between California and Louisiana cowbirds to specific factors, because the "Locality" variable contains all biotic and abiotic differences between the two locations. Geographic distance can be positively correlated with microbiota dissimilarity (*Dominguez-Bello & Blaser, 2011*), but local flora and fauna, photoperiod, available food, climate conditions, etc. may all affect microbiota and differ by locality. Environmental niche modeling on microbes/microbiota may indicate what abiotic environmental parameters are most important for shaping host-associated microbiota.

Diet and age were the next most frequently significant non-genetic variables. Dietary specialization is an important contributor to mammalian gut microbiota (*Colman, Toolson & Takacs-Vesbach, 2012*) and our results indicate this may also be true for birds (Fig. 4, Fig. S3, Table 3). Dietary classification was frequently significant across analyses but the actual contents of the stomach were not. How are these data compatible? In mammals, dietary specialization drives convergence in gut communities (*Muegge et al., 2011*) and once the community has stabilized, it is relatively immune to perturbation (*Walter & Ley, 2011*). This coincides very well with the apparent importance of diet but not of

actual stomach contents in cowbirds. Diet may be as important as locality in shaping the microbiota—when the Louisiana birds were analyzed alone, the diet variable became significant in 75% of the tests. Many birds had white millet in their stomachs, likely obtained by feeding at bird feeders; how human-supplied bird food (e.g., non-native seeds) impacts bird gut microbiota warrants further investigation. Furthermore, environmental feeding conditions (e.g., picking seeds off the ground, humidity, food decay, etc.) may also come into play.

Very young animals can have gut microbiota that is distinct from adults and they eventually undergo a transition period before reaching a stable, adult like community (e.g., *van Dongen et al., 2013*; *Vaishampayan et al., 2010*). Although the repeated significance of the age variable (Figs. 4A–4C) implies differentiation between younger and older individuals, adult cowbirds were no more similar to each other than they were to juveniles or hosts (Fig. S1) and an "adult cowbird" cluster was not observed in any analysis.

## Hypotheses

Across analyses and datasets, cowbirds and hosts were interspersed (Fig. 3)—neither cowbirds nor individuals from each brood host species clustered together. We therefore reject the Nature Hypothesis because there is no evidence for a specific cowbird gut community shared by all cowbirds. We are also unable to appropriately evaluate the Nurture Hypothesis, since an underlying assumption was that brood host species would be most similar to one another. The Environment Hypothesis is most strongly supported, since the locality variable is most frequently correlated with cowbird gut microbiota. Finally, we reject the Convergence Hypothesis, because adult cowbirds were no more similar to each other than they were to any other group.

Several aspects of these hypotheses warrant further discussion. The importance of locality and diet are consistent with an important role of ecology in shaping gut microbiota, so the Nurture Hypothesis remains a possible explanation of cowbird gut microbiota. For example, if the factors that shape gut microbiota are drawn from largely overlapping ecological niche space, a lack of bird taxonomic signal and generally low levels of clustering would be expected, in addition to the importance of parameters like locality and diet. An ideal experiment to test the Nurture Hypothesis would be to sample entire brood families from a single nest (parents, offspring, brood parasites) plus the cowbird mother.

Another caveat with the Nurture Hypothesis comes from the sampling, which included only samples that LSUMNS had previously collected. While some juveniles are most similar to brood host individuals, we have no way to evaluate whether that is because they were raised by that species. The Brown-headed Cowbird parasitizes approximately 250 bird species, only nine of which are represented here and of those sampled, there is an uneven distribution across age and locality (Table 1). This sampling is not ideal, however one priority of this study was to explore the amount of information gained through the collection and analysis of gut microbiota of museum specimens collected for other research purposes; therefore, we did not design our own sampling scheme so
much as utilize previously sacrificed animals. In other words, by taking advantage of recent technological (sequencing) advances, can we increase the amount of data gained from a given sampling effort? In the future, field biologists and microbial ecologists may work together to derive increasingly more data from museum collecting efforts and address interdisciplinary questions. Even modest microbiota analyses can provide the foundation for and preliminary answers to novel questions. For example, based on our results, targeted future studies should include as much taxonomic breadth as possible, with replicates from each locality. It was not anticipated that brood hosts would not cluster together, because they are genetically, ecologically and presumably environmentally most similar. We recommend including more than one individual from each species, when possible. Finally, regarding sampling, it is important to note that the single adult we have from California is a House Finch, which is parasitized by cowbirds but does not successfully rear their young (*Kozlovic, Knapton & Barlow, 1996*; *Lowther, 2013*). The inclusion of this cowbird "victim" has little affect on our interpretation of the data and could even be expected to be an outgroup to the other samples which are ecologically similar enough to raise cowbird chicks. However, House Finch does not appear significantly different from the other parasitized species. Its inclusion in the study has no effect on the major findings, that (1) cowbirds do not form their own cluster and (2) locality is the strongest correlate to cowbird gut microbiota.

Although we reject the Convergence Hypothesis, we leave room for a transition occurring sometime between hatching and fully-grown adult cowbirds. Furthermore, the scale of our analyses may have missed the critical transition period between young and adult microbiotas—which may vary by both bird and microbe species (*Moreno et al., 2003*; *Scupham, 2007*; *Scupham, 2009*)

High inter-individual variation appears to be a hallmark of microbiota studies (e.g., *Turnbaugh et al., 2010*; *Dethlefsen et al., 2006*; *Ruiz-Rodriguez et al., 2009b*), so much so that the concept of a "core microbiota" is in doubt (*Lozupone et al., 2012*). The birds in this study belong to a single order, the Passeriformes, and appear to have more variation than any of the non-avian orders we analyzed, with the exception of Hymenoptera (Fig. S3). Cowbirds in particular appear to have a highly variable gut microbiota, which may be a species level trait. The relative contribution of Proteobacteria and Firmicutes to cowbirds spans nearly all brood-host species and the cowbirds span nearly the entire dendrogram and NMDS plot (Fig. 3). However, the four Northern Cardinal samples appear to contain as much variation as the cowbirds (Figs. 2 and 3), so high variation in gut microbiota may be an order level trait.

Despite high levels of variation, taxonomic signals of the vertebrate hosts are frequently detected in microbiota studies (e.g., *Ochman et al., 2010*; *Dewar et al., 2013*; *Ley et al., 2008*), as are dietary classifications (*Ley et al., 2008*; *Anderson et al., 2012*). Our comparison of birds, mammals and insects revealed the taxonomic classes to be visibly and significantly different (Table 3, Fig. S3). Even at the order level, clustering was observed in all classes. No clustering by dietary classification was observed across or even within the classes (Fig. S3).

This study was conducted using a single marker and it relies on OTUs delimited from these genetic data. Metagenomic studies randomly sequence many loci across a microbiota sample and show that while taxonomic identity can vary widely across individuals, functional groups are highly conserved (*Lozupone et al., 2012*). The overlapping ecology, lack of taxonomic signal and significant effect of sampling locality indicate this would be an interesting application of metagenomics. Does the brood parasite's gut contain more functional categories than a traditional bird? What, if any, functional categories are most represented? Retesting all the above hypotheses with metagenomic data instead of fingerprint data would undoubtedly be valuable.

## ACKNOWLEDGEMENTS

We would like to acknowledge the LSU Museum of Natural Science Collection of Genetic Resources for providing samples for this study. Jacob Cooper, James Klarevas-Irby and Britt Perry helped prepare specimens. Thanks to Noah Reid and the Bird Lunch group at LSUMNS for fruitful discussion regarding this study. Caldwell Hahn, Tara Pelletier, Noah Reid, Jordan Satler, Matthieu Leray, Robert Toonen and several anonymous reviewers contributed helpful comments to the manuscript.

### Funding

The National Science Foundation (DEB-0956069 to BCC and RTB) and Sigma Xi Grants-in-Aid of Research (to SMH) funded this research. The funders had no role in study design, data collection and analysis, decision to publish, or preparation of the manuscript.

### Grant Disclosures

The following grant information was disclosed by the authors:
National Science Foundation: DEB-0956069.
Sigma Xi Grants-in-Aid of Research.

### Competing Interests

The authors declare no competing interests.

### Author Contributions

- Sarah M. Hird conceived and designed the experiments, performed the experiments, analyzed the data, contributed reagents/materials/analysis tools, wrote the paper, prepared figures and/or tables, reviewed drafts of the paper.
- Bryan C. Carstens, Steven W. Cardiff and Donna L. Dittmann contributed reagents/materials/analysis tools, reviewed drafts of the paper.
- Robb T. Brumfield conceived and designed the experiments, contributed reagents/materials/analysis tools, reviewed drafts of the paper.

**Peer**J

## Data Deposition

The following information was supplied regarding the deposition of related data:
Figshare: http://figshare.com/articles/Hird_Cowbird_fileset/957582.

## Supplemental Information

Supplemental information for this article can be found online at http://dx.doi.org/10.7717/peerj.321.

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

# Peer J

**Ochman H, Worobey M, Kuo C-H, Ndjango J-BN, Peeters M, Hahn BH, Hugenholtz P. 2010.** Evolutionary relationships of wild hominids recapitulated by gut microbial communities. *PLoS Biology* **8**:e1000546 DOI 10.1371/journal.pbio.1000546.

**Oksanen J, Blanchet FG, Kindt R, Legendre P, Minchin PR, O'Hara RB, Simpson GL, Solymos P, Stevens MHH, Wagner H. 2011.** *Vegan: community ecology package*, R package version 2.0-2. *Available at* http://CRAN.R-project.org/package=vegan.

**Ortega C. 1998.** *Cowbirds and other brood parasites.* Tucson: University of Arizona Press.

**Ortega C, Cruz A. 1992.** Differential growth patterns of nestling brown-headed cowbirds and yellow-headed blackbirds. *The Auk* **109**:368–376 DOI 10.2307/4088205.

**Price MN, Dehal PS, Arkin AP. 2009.** FastTree: computing large minimum-evolution trees with profiles instead of a distance matrix. *Molecular Biology and Evolution* **26**:1641–1650 DOI 10.1093/molbev/msp077.

**Quinn G, Keough M. 2002.** *Experimental design and data analysis for biologists.* Cambridge: Cambridge University Press.

**R Development Core Team. 2010.** R: a language for statistical computing. R Foundation for Statistical Computing, Vienna, Austria.

**Ruiz-Rodriguez M, Valdivia E, Soler JJ, Martin-Vivaldi M, Martin-Platero AM, Martinez-Bueno M. 2009a.** Symbiotic bacteria living in the hoopoe's uropygial gland prevent feather degradation. *Journal of Experimental Biology* **212**:3621–3626 DOI 10.1242/jeb.031336.

**Ruiz-Rodriguez M, Lucas FS, Heeb P, Soler JJ. 2009b.** Differences in intestinal microbiota between avian brood parasites and their hosts. *Biological Journal of the Linnean Society* **96**:406–414 DOI 10.1111/j.1095-8312.2008.01127.x.

**Schloss PD, Handelsman J. 2005.** Introducing DOTUR, a computer program for defining operational taxonomic units and estimating species richness. *Applied and Environmental Microbiology* **71**:1501–1506 DOI 10.1128/AEM.71.3.1501-1506.2005.

**Schloss PD, Westcott SL, Ryabin T, Hall JR, Hartmann M, Hollister EB, Lesniewski RA, Oakley BB, Parks DH, Robinson CJ, Sahl JW, Stres B, Thallinger GG, Van Horn DJ, Weber CF. 2009.** Introducing mothur: open-source, platform-independent, community-supported software for describing and comparing microbial communities. *Applied and Environmental Microbiology* **75**:7537–7541 DOI 10.1128/AEM.01541-09.

**Scupham A. 2007.** Succession in the intestinal microbiota of preadolescent turkeys. *FEMS Microbiology Ecology* **60**:136–147 DOI 10.1111/j.1574-6941.2006.00245.x.

**Scupham A. 2009.** Campylobacter colonization of the turkey intestine in the context of microbial community development. *Applied and Environmental Microbiology* **75**:3564–3571 DOI 10.1128/AEM.01409-08.

**Sekirov I, Russell SL, Antunes LCM, Finlay BB. 2010.** Gut microbiota in health and disease. *Physiological Reviews* **90**:859–904 DOI 10.1152/physrev.00045.2009.

**Sharon G, Segal D, Ringo JM, Hefetz A, Zilber-Rosenberg I, Rosenberg E. 2010.** Commensal bacteria play a role in mating preference of Drosophila melanogaster. *Proceedings of the National Academy of Sciences of the United States of America* **107**:20051–20056 DOI 10.1073/pnas.1009906107.

**Spaw CD, Rohwer S. 1987.** A comparative study of eggshell thickness in cowbirds and other passerines. *The Condor* **89**:307–318 DOI 10.2307/1368483.

**Turnbaugh PJ, Quince C, Faith JJ, Mchardy AC, Yatsunenko T, Niazi F, Affourtit J, Egholm M, Henrissat B, Knight R, Gordon JI. 2010.** Organismal, genetic, and transcriptional variation in the deeply sequenced gut microbiomes of identical twins. *Proceedings of the National Academy of Sciences of the United States of America* **107**:7503–7508 DOI 10.1073/pnas.1002355107.

**Vaishampayan PA, Kuehl JV, Froula JL, Morgan JL, Ochman H, Francino MP. 2010.** Comparative metagenomics and population dynamics of the gut microbiota in mother and infant. *Genome Biology and Evolution* **2**:53–66 DOI 10.1093/gbe/evp057.

**van Dongen W, White J, Brandl H, Moodley Y, Merkling T, Leclaire S, Blanchard P, Danchin É, Hatch S, Wagner R. 2013.** Age-related differences in the cloacal microbiota of a wild bird species. *BMC Ecology* **13**:11 DOI 10.1186/1472-6785-13-11.

**Walter J, Ley R. 2011.** The human gut microbiome: ecology and recent evolutionary changes. *Annual Review of Microbiology* **65**:411–429 DOI 10.1146/annurev-micro-090110-102830.

**Wang Q, Garrity G, Tiedje JM, Cole JR. 2007.** Naive Bayesian classifier for rapid assignment of rRNA sequences into the new bacterial taxonomy. *Applied and Environmental Microbiology* **73**:5261–5267 DOI 10.1128/AEM.00062-07.

**Woolfenden BE, Gibbs HL, Sealy SG, McMaster DG. 2003.** Host use and fecundity of individual female brown-headed cowbirds. *Animal Behaviour* **66**:95–106 DOI 10.1006/anbe.2003.2181.

**Zilber-Rosenberg I, Rosenberg E. 2008.** Role of microorganisms in the evolution of animals and plants: the hologenome theory of evolution. *FEMS Microbiology Reviews* **32**:723–735 DOI 10.1111/j.1574-6976.2008.00123.x.

**Zoetendal EG, Akkermans ADL, Akkermans-van Vliet WM, de Visser JAGM, de Vos WM. 2001.** The host genotype affects the bacterial community in the human gastrointestinal tract. *Microbial Ecology in Health and Disease* **13**:129–134 DOI 10.1080/089106001750462669.