# Peer review of "Sampling locality is more detectable than taxonomy or ecology in the gut microbiota of the brood-parasitic Brown-headed Cowbird (Molothrus ater)"

_PeerJ, doi:10.7717/peerj.321_

## Round 0.1 · original submission · Major Revisions

I am sorry for the delay in turning this manuscript around, but I had 8 consecutive referees decline to review the paper, and it took a while to get the reviews completed. Your manuscript was reviewed by referees familiar with each of the avian biology and molecular aspects of the work, but these referees had very different impressions of the work. The conclusion of the molecular ecologist was that the work was sound and that you have adequately discussed the limitations of the study due to the sampling strategy. The avian biologist disagrees strongly and felt that the work was fundamentally unpublishable as a result of the sampling strategy and some errors in the manuscript which made it impossible to draw meaningful conclusions from the data.
Having read the paper myself, I can see both sides of that disagreement – cowbirds do lay eggs in house finch nests, but based on the reference provided by the referee fail to rear any chicks from those nests. I am not an avian biologist, but certainly the issue of inclusion of the house finch as the sole potential host from CA needs to be resolved explicitly in the manuscript. In general, I am loathe to instruct people to go back and produce additional data, but in this case, as the referee points out, one solution may be to add an important host such as the redwing black bird to replace this critical sample in the analyses. I leave it to the authors as to how they decide to resolve this issue, but it appears the key point in the evaluation of suitability for this referee.
To be blunt, I am still undecided whether the paper is acceptable here or not, and would like to see the authors address the comments and revise the manuscript for re-review. My impressions is that neither of these two referees are likely to change their mind about the suitability for acceptance, and so I will select a third referee to review the manuscript after revisions. In this way I can ask a second opinion on the issues of the avian biology at the same time as see for myself how the authors respond to this criticism without delaying the manuscript further. I can imagine a revision to correct the issues raised by the second referee and appropriately tempered conclusions might well result in a publishable manuscript, but I make no assurances in this regard - the paper will be re-reviewed by a third referee before I make my final decision.

Reviewer 1 ·

Basic reporting

This paper examines factors that determine the gut microbiota of the parasitic Brown-headed Cowbird. This is an interesting idea, but unfortunately it’s poorly designed and poorly written.

Lines 26-33: no scientific names given for these species, but more importantly why are these species relevant to your study? It appears you’ve decided to mention a few random species.

Line 38: It’s atypical for young birds being referred to as “babies”

Cowbird host numbers throughout should be updated. See Lowther's data http://fieldmuseum.org/sites/default/files/Molothrus-02nov2012.pdf

Line 39: Cowbird eggs are "not camouflaged"? I assume you mean mimetic, which is very different because I’d argue that cowbird eggs are camouflaged.

44: Cowbirds are not "host tolerant". See Peer et al. 2013 Chinese Birds 4:15-30.

85: I’m unfamiliar with the term “brood host species”

Line 63: Mourning Doves regurgitate but young place mouth inside that of adults.

Lines 264-266: reference?

Figure 1 caption: “cartoons”?

Experimental design

The experimental design is weak and appears to have been based on what samples were available in the freezer. Most notable is the fact they've chosen an unsuitable host species, the House Finch. Please see Kozlovic et al. 1994 Condor 98:253-258. House Finches are unsuitable hosts; they feed their young seeds and this is not a suitable diet for cowbirds. This single host sample represents their entire sample of hosts from California.

Additionally, the sample size for hosts is extremely small, the largest from a single host being 4. Why not focus on obtaining larger samples for several important hosts? Why not choose hosts that are closely related to cowbirds such as a Red-winged Blackbird. This would allow you to determine whether there are specific differences between cowbirds and their close relatives that make them more effective brood parasites.

Validity of the findings

I agree with the authors that their sampling was "not ideal" (line 327) and they made this statement based on the incorrect assumption that the House Finch is a suitable host species. Based on the design and small samples there is very little that can be concluded from this study.

·

Basic reporting

No comments

Experimental design

This paper tests whether microbial communities in the guts of brood parasitic cowbirds are structured by cowbird genetics or extrinsic factors such as locality, diet or brood host. The context and working hypothesis are well presented. Authors targeted the standard 16S region and employed analytic tools provided in the QIIME package to test their hypothesis. Lab procedures and analysis are also well presented.

Authors used guts from museum specimen for the study. I understand their justification to do so. However, more information should be provided in the method section about the specimen collection (i.e. collection methods, collection dates, collector, time before specimen was frozen…) to convince the reader that these factors do not affect the results. This additional information could eventually be used to quantify how much of the variation in gut microbiota community structure is driven by differences in the collection procedure.

Validity of the findings

Results presented in this paper show no support for the nature hypothesis while sampling locality, diet and age drive differences in gut microbiota community structure. Authors put their data into a broader context by comparing microbial communities to mammals and insects. The limitations of the study due to the sampling strategy are appropriately discussed.

Additional comments

No comments

---

## Round 0.2 · accepted · Accept

Having read through your reply and looking at the extensive revisions to your manuscript, I feel that you have addressed the referee concerns to my satisfaction. In particular, your revision of the text to address the criticisms of referee #1 regarding the single victim/host from CA is well justified and convincing to the additional reader I asked to examine the paper.

Rather than request a new review and potential additional changes that would delay your decision, I asked a colleague who had not seen the original submission to read the revised manuscript and tell me if they were convinced or not before I decided whether to send it out for formal review. Their feedback was that they felt the conclusions of the paper were in line with the data, and those conclusions did not hinge on any individual sample in the study, and I am inclined to agree. Thus, I feel that your extensive revisions to both the introduction and discussion are sufficient to merit acceptance of the revised manuscript.

A final note is that while unnecessary, I appreciate the acknowledgement in your manuscript for the feedback provided by both the referee and editor. Yours is the first manuscript I have handled as editor in which I have been acknowledged, and it is a pleasant surprise that the time and effort invested by the editor is still appreciated by some. Thank you.